A machine learning-based hybrid recommender framework for smart medical systems

Wei Jianhua 1
Yan Honglin 2
Shao Xiaoli 3
Zhao Lili 4
Han Lin 3
Yan Peng 1
Wang Shengyu 5 wangshengyu1065@xiyi.edu.cn
1 The Bidding Procurement Office, The First Affiliated Hospital of Xi’an Medical University , Xian , China
2 Department of Gastroenterology, The First Affiliated Hospital of Xi’an Medical University , Xian , China
3 Hospital Evaluation and Accreditation Office, The First Affiliated Hospital of Xi’an Medical University , Xian , China
4 Department of Scientific Research, The First Affiliated Hospital of Xi’an Medical University , Xi’an , China
5 Pulmonary and Critical Care Medicine, The First Affiliated Hospital of Xi’an Medical University , Xi’an , China
Asif Muhammad
Electronic publication date: 2024 Feb 20
Publication date: 2024
Volume: 10
Electronic Location ID: e1880
Received 2023 Dec 7; Accepted 2024 Jan 24
Copyright: © 2024 Wei et al.
Copyright year: 2024
Copyright holder: Wei et al.
License: This is an open access article distributed under the terms of the Creative Commons Attribution License, which permits unrestricted use, distribution, reproduction and adaptation in any medium and for any purpose provided that it is properly attributed. For attribution, the original author(s), title, publication source (PeerJ Computer Science) and either DOI or URL of the article must be cited.
License URL: https://creativecommons.org/licenses/by/4.0/

Keywords: Medical registration, Medical evaluation, Big data, Deep learning algorithms, Doctor recommendations

Funding: The authors received no funding for this work.

==============================
This article presents a hybrid recommender framework for smart medical systems by introducing two methods to improve service level evaluations and doctor recommendations for patients. The first method uses big data techniques and deep learning algorithms to develop a registration review system in medical institutions. This system outperforms conventional evaluation methods, thus achieving higher accuracy. The second method implements the term frequency and inverse document frequency (TF-IDF) algorithm to construct a model based on the patient’s symptom vector space, incorporating score weighting, modified cosine similarity, and K-means clustering. Then, the alternating least squares (ALS) matrix decomposition and user collaborative filtering algorithm are applied to calculate patients’ predicted scores for doctors and recommend top-performing doctors. Experimental results show significant improvements in metrics called precision and recall rates compared to conventional methods, making the proposed approach a practical solution for department triage and doctor recommendation in medical appointment platforms.

Introduction

Assessing the service level of medical institutions is vital to enhancing patient care and promoting quality improvements (Lin, Shih & Ho, 2023; Xu & Zhang, 2022). The conventional approach to assigning grades to medical institutions based on assessment scores involves using mainstream Software called a SaaS system. To do this, various types of data from each department are collected, the predetermined processing methods are employed to analyze, and thus a label used for the assessment of the service of the hospital is generated via quantitative integration (Mills, Kwakkenbos & Carrier, 2018). However, this evaluation process inaccurately reflects the relationship between indicators in each department during the review and neglects the linkage relationship among various departments. Consequently, inaccurate service level evaluations are generated. To address these limitations, an optimized medical institution registration and review scheme is required.

Recent years have witnessed the rising popularity of the implementation of big data, deep learning, and neural networks that provide remarkable performance increases in crunching data across diverse fields ranging from computer vision to natural language processing and text signal processing (Meguid & Collins, 2017; Abdalla et al., 2020; Alkhalil et al., 2021; Kim et al., 2022). These technologies have surpassed the level of human performance in areas such as image classification, object detection, semantic segmentation, and text translation.

Autonomous registration has high error rates due to imperfect medical knowledge systems. So, manual triage tables are generally utilized in hospitals but they do not suffice to efficiently triage large numbers of patients and thus cause long waiting times, negatively impacting patients’ experience and contributing to doctor-patient conflicts (Hardey, 1999). To reduce the burden on desks manually doing triages, many large hospitals have set up online registration and triage functionalities without fundamentally addressing consulting power issues. Medical service institutions utilizing the Internet integrate information from various hospitals to provide patients with online registration or direct consultation services. However, these platforms still heavily rely on manual input and lack satisfying triage and doctor recommendation functionalities. As big data-based artificial intelligence technology advances, experts have been researching how to reform conventional medical services, improve patients’ medical experience, and enhance doctor-patient relationships. Hence, intelligent triage systems allow patients to choose doctors at departments independently and get information at which to take appointments regarding their health issues, reducing the cost of manual triage services. Furthermore, Internet-based medical platforms may combine conventional hospital triage platforms with collaborative filtering recommendation algorithms to realize an intelligent department triage and doctor recommendation system concurrently, although this approach also needs to deal with the exponential growth of medical information, leading to slow recommendation processes and difficulties in handling cold start problems (Powell, Inglis & Ronnie, 2011).

Hence, the motivation behind the proposed hybrid recommender framework for smart medical systems is to address the challenges faced by medical institutions and patients in the context of service-level evaluations and physician recommendations in medical appointment platforms since conventional evaluation approaches often lack accuracy, leading to suboptimal service quality. Furthermore, conventional recommendation systems may not effectively match patients with the most suitable doctors.

The first method presented in the manuscript addresses the need for accurate service-level evaluations. By leveraging big data techniques and deep learning algorithms, the registration review system of the proposed method improves the accuracy of evaluations by employing advanced encoding schemes, feature extraction, and classification techniques. The proposed approach significantly outperforms conventional methods, providing medical institutions with more precise and reliable evaluations regarding their service levels.

The second method focuses on enhancing doctor recommendations for patients. By constructing a model used for patients’ symptoms a vector space based on the TF-IDF algorithm and incorporating score weighting is employed. Additionally, the introduction of modified cosine similarity, K-means clustering, alternating least squares (ALS) matrix decomposition, and user collaborative filtering algorithms further enhances the recommendation process.

The experimental results demonstrate substantial improvements in precision and recall rates when compared to conventional approaches, offering patients more accurate and personalized doctor recommendations.

The contributions of the research are as follows: Firstly, the proposed hybrid recommender framework provides a comprehensive solution to address the challenges of service level evaluations and doctor recommendations in medical appointment platforms. Integrating big data techniques, deep learning algorithms, and innovative methodologies improves the accuracy and efficiency of both evaluation and recommendation systems.

Secondly, service level evaluations are substantially enhanced when compared to conventional methods. By leveraging advanced encoding, feature extraction, and classification techniques, it generates more precise and reliable service-level labels for medical institutions. This enables them to better understand their performance and make informed decisions for improvement.

Finally, the improved doctor recommendation approach enhances the patient experience by accurately matching patients with the most suitable doctors. By considering patients’ symptoms, employing advanced similarity measures, and utilizing collaborative filtering algorithms, the system provides personalized recommendations that significantly outperform conventional methods. This ensures that patients receive appropriate medical care, leading to improved healthcare outcomes (Liu & Zhao, 2023).

The rest of the article is structured into four main sections. “The Evaluation System of Big data-based Registration” focuses on the development of a big data-based evaluation system for patient registrations that elaborates on deep learning algorithms to improve the accuracy of service-level evaluations for medical institutions. The section describes the implementation of a context encoder, multi-scale neighborhood feature extraction, and maximum value-based eigenvalue correction to encode multiple index data and refine extracted features. It also explains how a convolutional neural network (CNN) model is employed to classify and generate results for service-level labels and highlights the higher accuracy achieved by the proposed system. A comparison is made with conventional evaluation methods. “Hybrid Recommendation Algorithm-based Intelligent Triage System” introduces the hybrid recommendation algorithm-based intelligent triage system that constructs a vector space model representing patients’ symptoms by employing the TF-IDF algorithm, along with the incorporation of score weighting to enhance its performance. Also, the use of modified cosine similarity to predict visiting departments and the application of an improved trust relationship-based K-means is defined to cluster patients. Furthermore, it outlines the utilization of ALS matrix decomposition and user collaborative filtering algorithm to calculate patients’ predicted scores for doctors and recommend a list of doctors with top scores. Experimental results demonstrating significant enhancements in precision and recall rates are presented, highlighting the effectiveness of the system in department triage and doctor recommendations concurrently. “Discussion” discusses the advantages and disadvantages of the proposed method. Finally, “Conclusion” concludes the article by summarizing the key findings and contributions. It emphasizes the improvements in accuracy and efficiency achieved in service level evaluations and doctor recommendations, respectively. Moreover, the practical implications of the system are discussed, offering an effective solution to enhance the efficiency and accuracy of medical appointment platforms. Potential directions for future research are presented, providing insights for further advancements in smart medical systems.

The evaluation system of big data-based registration

An evaluation system dealing with big data-based registration is a software application or platform that leverages big data analytics techniques to assess registration processes. It is commonly implemented in institutions or large organizations like medical institutions that require efficient and accurate registration systems.

Composition and workflow of the proposed system

Figure 1 shows a schematic diagram of the proposed system used by medical institutions’ registration review systems.

Figure 1 Schematic diagram of the composition of the proposed system.

The system comprises various modules that work cohesively to assess the performance of each department. The first module is the collection of index data coded (110), which gathers multiple index data for each department. These data pass through a context encoder with an embedding layer to obtain semantic feature vectors. The multi-scale feature extraction module coded (130) cascades the semantic feature vectors into department feature vectors and employs multi-scale neighborhood feature extraction to attain multiple department feature vectors. Next, the feature vector correction coded (140) performs maximum value-based eigenvalue correction on the multi-scale department feature vectors to obtain corrected multi-scale departmental feature vectors. The inter-department association encoding coded (150) arranges these corrected feature vectors into a two-dimensional feature matrix and passes it through a CNN extracting features to obtain a classification feature map. Finally, the review result generation coded (160) implements a classifier to generate classification results that represent the service level label of the medical institution.

The goal of the research is to evaluate the service level of a medical institution by characterizing the correlation between indicators in each department and the relationship between departments. The current evaluation method relies on mainstream SaaS system software, which collects index data from each department for quantitative integration and evaluation by employing predetermined processing methods to determine hospital service ratings. However, this approach fails to account for the relationships between indicators within departments or between different departments, resulting in inaccurate grades.

To address these limitations, the research proposes a technical scheme that employs deep neural networks (DNNs) to extract high-dimensional hidden features associated with the linkage between indicators and departments from multiple index data collected from each department. These features are then fed into a classifier to evaluate the service level of the medical institution.

Department indicators include personnel, funds, scientific research achievements, and service attitudes. The index data encoding module 120 of each department passes multiple index data through a context encoder, including an embedding layer, to obtain multiple semantic feature vectors. The context semantic encoder is utilized to perform contextual semantic encoding to extract correlations between index data items. Multiple index data items are considered a text sequence, and a context encoder is employed to encode all data items based on global context semantics to attain multiple semantic feature vectors.

Figure 2 shows the workflow of the proposed registration review method that comprises several stages, including S110: Acquiring index data from each department of the medical institution under review. S120: Using a context encoder to process the index data and generate multiple index semantic feature vectors. S130: Concatenating the multiple index semantic feature vectors into department feature vectors. S140: Performing eigenvalue correction based on the maximum value on the multi-scale department feature vectors of each department to obtain corrected multi-scale department feature vectors for each department. S150: Arranging the corrected multi-scale department feature vectors into a two-dimensional feature matrix and employing a CNN model to generate a feature map for classification. S160: Passing the classification feature map through the classifier to obtain the classification result, which represents the service level label of the medical institution under review.

Figure 2 Workflow of the proposed system.

Figure 3 depicts the framework of the proposed system, which has multiple modules to evaluate the service level of medical institutions accurately. First, the system acquires index data of each department of the medical institution to be reviewed. Next, it utilizes a contextual encoder with an embedding layer to generate semantic feature vectors from the index data. These semantic feature vectors are then concatenated into department feature vectors and processed by employing multi-scale neighborhood feature extraction modules to obtain multi-scale department feature vectors corresponding to each department. The multi-scale department feature vectors undergo eigenvalue correction based on the maximum value to obtain corrected multi-scale departmental feature vectors.

Figure 3 Working framework of the proposed system.

Finally, these corrected multi-scale departmental feature vectors are arranged into a two-dimensional feature matrix and passed through a CNN. The feature map is fed into a classifier to obtain service-level labels for the medical institution review.

Internal function modules

Internal function modules are individual components within a software system that perform specific functions. These modules work together to enable the overall functioning of the system.

Encoding module

The index data encoding module 120 comprises two main units: the embedding vectorization unit 121 and the context encoding unit 122, respectively. The embedding vectorization unit 121 converts multiple index data into an embedding vector sequence by employing the embedding layer of the context encoder. The context encoding unit 122 implements the converter-based Bert model to perform semantic encoding on global-based context by using the sequence of embedding vectors, generating multiple semantic feature vectors that capture the correlation between different index data items in each department (Liu & Zhao, 2023). The proposed index data coding module performs contextual semantic encoding on index data by implementing DLMs to extract high-dimensional hidden features associated with the correlation between different index data items.

The multi-scale feature extraction module 130 concatenates multiple index semantic feature vectors into department feature vectors and passes them through the multi-scale neighborhood feature extraction module. So, multi-scale department feature vectors are attained corresponding to each department. Then, CNN is implemented for local feature extraction, where the convolution kernel moves along the time dimension in the form of a sliding window and outputs a weighted sum of the data within each time-series segment (Alpaslan & Hanbay, 2020). Large and small-scale convolution kernels are combined to extract features of different timing scales, achieving smooth input data, and minimizing information loss.

Suppose that the size of the input data is denoted by (n, t, d), where n, t, and d represent the sample size, time series length, and the characteristic dimension of each time step, respectively. In CNN, k convolution nuclei of different sizes are usually employed to extract multi-scale neighborhood features, where the size of the ith convolution kernel is denoted by (hi, d, ci), where hi and ci represent the window size, and the number of the convolution kernel, respectively. Then, for a convolution kernel, its weight, bias, and output vector can be defined as W(i)=Rhi⋅d⋅ci,b(i)=Rci,andZ(i)=Rn(t−hi+1)ci, respectively. The convolution operation is conducted by the convolution checking on the input data. Equation (1) represents this relation.

(1) Z=f(∑j=1d⁡wj(i)∗xj+b(i))

where * and f stand for convolution operation and activation function, respectively. The computed output of the convolution kernel is a tensor represented by (n, t − h + 1, ci) magnitude, where each time step corresponds to a ci dimensional eigenvector. Often, pooling is also required to reduce feature dimensions and model complexity. The multiple convolution kernels of output can be pieced together, forming a (n, t − h + 1, ∑i=1k⁡ci) size of a tensor, among them, the h=max{ℎ1,…,ℎk} denotes the maximum window size in the CNN model. This tensor is the multi-scale neighborhood feature, which can be directly employed in the subsequent classification task.

To sum up, mathematical expressions are required to describe the CNN model and the specific implementation process of its convolution operation, including relevant parameters such as weight matrix, bias vector, and activation function, and multi-scale neighborhood features extracted through convolution operation and pooling operation, respectively.

Feature extraction module

Figure 4 shows the block diagram of the multi-scale feature extraction module 130 in the registration review system of the medical institution.

Figure 4 Block diagram of the feature extraction module of the proposed system.

The module comprises several units: concatenation unit 131, first neighborhood scale encoding unit 132, second neighborhood scale encoding unit 133, and multi-scale feature cascading unit 134. Concatenation unit 131 combines multiple index semantic feature vectors to obtain department feature vectors. First neighborhood scale encoding unit 132 applies one-dimensional convolution processing with a kernel length on the department feature vector to derive the first scale department feature vector. Second neighborhood scale encoding unit 133 applies one-dimensional convolution processing with a different kernel length on the department feature vector to generate the second scale department feature vector. Lastly, multi-scale feature cascading unit 134 concatenates the first and second-scale department feature vectors to produce multi-scale department feature vectors for each department. These feature vectors capture the relationship between different index data at different scales.

The eigenvector correction module 140 (ECM) performs eigenvalue corrections based on multi-scale department feature vectors, generating corrected vectors for a better representation of medical institution service levels. Thus, corrected vectors are arranged in a two-dimensional feature matrix and utilized as a feature extractor for a CNN model. However, coding fluctuations of index-related semantic features at multiple scales may cause phase differences between corresponding department feature vectors. Such differences can affect the classification accuracy of the feature map by hindering proper aggregation of the matrix. To address this issue, the system aligns multi-scale department feature vectors by implementing advanced techniques before arranging them into a feature matrix. With aligned vectors, related features occupy consistent positions within the matrix, improving the aggregation effect on the class probability of the classifier and boosting classification accuracy.

Specifically, the eigenvector correction module 140 is further configured to: employ Eq. (2) to perform eigenvalue correction based on the maximum value on the multi-scale department eigenvectors that correspond to each department to obtain the corrected multi-scale department features relevant to each department vector. Equation (2) gives this vector.

(2) V′=V⊙e−sin(2π⊙V⊙vmax−1)

where V′, V, and vmax represent the multi-scale department feature vector, the corrected multi-scale department feature vector, and the maximum eigenvalue of the multi-scale department feature vector, respectively.

The proposed method employs the aggregation of the wave function representation to address the challenge of phase differences between multi-scale department feature vectors. It employs amplitude to represent intensity information and phase knowledge to characterize periodic position information, enabling complex vector information representations. This mitigates negative impacts on class probability aggregation, leveraging in-phase enhancement and out-of-phase cancellation based on the wave function to enhance classification accuracy. The approach improves the rationality and accuracy of the service level evaluations of medical institutions. Overall, the system leverages the DLMS and the aggregation of the wave function representation for more accurate and reliable service-level evaluations of medical institutions.

Associative encoding module

The research employs the inter-department association encoding module 150 to arrange department feature vectors into a 2D matrix. The CNN model is then employed to extract classification features. Correlation characteristics are extracted within each department. To consider the linkage relationships between departments, the corrected multi-scale department feature vectors are arranged into a 2D matrix and CNN is implemented to attain classification feature maps, extracting associated features between departments.

The inter-department association encoding module 150 is also employed for each CNN layer to perform the forward passes of input data. A convolution unit performs 2D convolution processing by implementing the convolution kernel. Then, a pooling unit performs mean pooling on the convolutional feature map based on the local feature matrix. An activation unit performs nonlinear activation on the pooled feature map to obtain the activation feature map. The output of the last CNN layer is a classification feature map.

The whole process can be expressed by Eq. (3).

(3) Fout=CNN(Fin)

where Fin denotes the input data, CNN represents the convolutional neural network model, and Fout characterizes the output feature map. In this process, the CNN model converts the two-dimensional feature matrix into the classification feature map through multi-layer convolution, pooling, and activation operations, respectively, to realize the classification diagnosis of patients.

Equation (4) represents the specific formula for each step in the inter-department association coding module 150.

(1) For each department, a multi-scale feature extractor was utilized to encode its indicator data and extract the correlation features among indicators. The feature graph X∈RH×W×Cin was input into a multi-scale feature extractor to obtain feature graph {Xs}s=1s with different scales, where S represents the number of scales. Equation (4) presents the feature map of each scale.

(4) Xs=fs(X),s=1,2,…,S

where fs(·) represents the eigenfunction employed to derive the s scale.

(2) The corrected multi-scale department feature vectors corresponding to each department were arranged into a two-dimensional feature matrix, which contained the correlation features among the internal indicators of each department and the indicators of each department. For each department, its corresponding multi-scale feature vectors {Xk}k=1ki were arranged into a two-dimensional feature matrix according to certain rules Mi∈RH×W×C, where Ki represents the number of samples in the i-th department, C = Cin × S represents the length of feature vector at each position. Equation (5) presents the process.

(5) Mi=(h,w,c)={xks(h′,w′,c′),if(h′,w′,c′)∈P(h,w,c)0,otherwise

(h′, w′, s′, c′) is an element of the eigenvector Xsk and P(h,w,c) is a set of all of the elements that need to extract (h′, w′, s′, c′) collection.

(3) The CNN model was employed to process the input data and obtain the output feature map. The two-dimensional feature matrix Mi was input into the CNN and the classified feature graph Fi∈RHo×Wo×Co was obtained, where Ho, Wo, and Co represent the height, width, and number of channels of the output feature graph, respectively. Equation (6) represents the CNN.

(6) Fi=CNN(Mi)

where the CNN(·) represents the convolutional neural network model, and Fi denotes the output feature map.

(4) The output of the last layer was the classification feature map, which was implemented to classify patients. The classification feature graph {Fi}Ni = 1 of all departments was summarized to obtain the final classification feature graph Fout ∈ RHo×Wo×N, where N represents the number of departments. Equation (7) represents this process.

(7) Fout(h,w,i)=Fi(h,w),i=1,2,…,N

where (h,w) denotes a position in the output feature graph. Finally, the classification algorithm can be employed to classify the classification feature map, to realize the classification diagnosis of patients.

Classification module

The result generation module of reviews (160) is employed to pass the classification feature map through a classifier to obtain a classification result that is utilized to represent the service level label of the medical institution to be reviewed.

The result generation module of evaluations (160) is further employed to process the classification feature vector presented in Eq. (8) to obtain the classification result by employing the classification feature map.

(8) O=f(W,b)=[∑i=1k⁡exp(Wix+bi)]−1[exp(W1x+b1)exp(W2x+b2)...exp(Wkx+bk)]

where O, Wi, and bi denote the output matrix, and the weight and bias matrices corresponding to the ith classification, respectively.

Other functional modules

The registration review system for a medical institution has several functional modules, including a login module, a SaaS background management module, an ICD-10 dictionary unit, an ICD-9-cm-dictionary unit, an M-code dictionary unit, an implementation plan module, and a data collection module. For example, the SaaS background management module is further divided into an organization management unit, a personnel management unit, an authority management unit, an ICD-10 dictionary unit, an ICD-9-cm-dictionary unit, and an M code Dictionary unit, a review standard unit. These modules allow for the management of institutional users, adding and deleting personnel, assigning permissions, retrieving national standards, and reviewing criteria to support medical institutions’ configuration needs. The evaluation standard module provides a summary of collected and reported data and displays the rating rules and scoring method. The interpretation of the detailed rule module allows hospital administrators to interpret national standards and display the detailed rules for each clause. The implementation plan module allows hospital administrators to formulate data collection plans based on the review terms, while the data acquisition module allows staff to perform periodic reporting operations according to assigned tasks. After data collection is finished, hospital managers can track the reporting status and calculate completion and accuracy rates.

Internal electronics

The electronic device has processors and memory to execute computer programs for evaluating medical institutions’ service levels based on index data. The memory includes volatile and non-volatile storage media that store the program’s products. The processor controls components and processes instructions and data from various departments. So, the data can be stored for further processing (Kawahara, Ito & Takemura, 2012).

Hybrid recommendation algorithm-based intelligent triage system

A hybrid recommendation algorithm utilizing an intelligent triage system is a software that utilizes a combination of recommendation algorithms and intelligent triage techniques to assist in the process of prioritizing and categorizing incoming requests, tasks, or issues. It aims to optimize resource allocation, streamline workflow, and improve efficiency in handling incoming workload.

Construction of symptom vector space model

TF-IDF algorithm-based patient symptom model

To construct a patient symptom model based on the TF-IDF algorithm, medical treatment samples from the data warehouse of the medical company can be employed. However, words used for different symptom segmentations have varied importance in describing the real condition of patients. Therefore, it is necessary to assign weights to a word describing each symptom to measure its importance in a patient’s condition. To determine the significance of a symptom participle in a patient’s chief complaint, its frequency in the patient’s complaint must be compared to its frequency in other chief complaints. A higher frequency of a symptom participle in a patient’s complaint and a lower frequency in other complaints indicates that the symptom is more important in the patient’s current symptoms.

Different types of symptom segments in the main complaint can be utilized as featured words. Then, the TF-IDF of various symptom segments of patients can be utilized as the weight of the featured words to generate the multidimensional space vector model (VSM). The construction of the patient symptom entity is thus completed by collecting those frequencies. TF-IDF algorithm quantifies the importance of a symptom participle in the patient’s chief complaint by calculating the ratio of the frequency of a symptom participle in the patient’s chief complaint to the total frequency of all symptom participles in the patient’s current chief complaint. Equation (9) presents the calculation.

(9) TFi,j=ni,j/n*,j

where ni,j denotes the frequency of symptom participle ti appearing in patient j’s current complaint, n*,j represents the total number of occurrences of all symptom participle in patient j’s current complaint. IDF represents the inverse document frequency of symptom participle i in the patient j’s main complaint file. Equation (10) presents the IDF calculation.

(10) IDFi=log(N/Di+1)

The condition of patient j can be represented by the VSM v(j) composed of n symptom featured word segments in the chief complaint of patient j. Equation (11) presents it.

(11) v(j)=(Wj1,Wj2,Wj3,…,Wjn)

Score-weighted TF-IDF-based department symptom vector

To accurately measure the value of different symptom participles for department-featured descriptions, the TF-IDF algorithm introduced earlier is implemented. However, when constructing an individual department symptom vector model, some differences occur. The patient’s symptom participle is based on the patient’s initiative expression, which is closer to the actual situation. In contrast, the latter is based on the frequency of occurrence of each symptom within the patient’s medical record. Regarding departmental triage, conventional triage methods may have erroneously assigned departments in the past, leading to misdiagnosis and incorrect treatment issues. However, by considering patients’ feedback and ratings of doctors, we can infer the suitability of a department for treating certain symptoms. If a patient visits a particular department and rates the doctor with a high score, the visited department is likely more suitable for treating the patient’s symptoms. This also implies that the set of the patient’s symptoms has a higher contribution to this department’s feature description. Conversely, a lower score indicates that the patient’s symptoms contribute less to the department’s description.

Therefore, to improve the accuracy of department triage, we need to improve the conventional TF-IDF algorithm and add the weight effect of scoring. Equation (12) presents the calculation of the score-weighted word frequency TF of symptom i to a certain department. Equation (12) presents its calculation.

(12) TFi=∑k∈U⁡ni,k⋅rk∑k∈U⁡n∗,k⋅rk

where ni,k represents the frequency of symptom i in the chief complaint symptom set of patient k in department d, and rk (1 ≤ rk ≤ 5) represents the score given by the patient to the doctor after the visit is realized. N*,k represents the total number of occurrences of all symptom words in the chief complaint symptom set of patient k. U denotes the collection of all patients in department d. The department d can be represented by the contribution of its n symptoms forming an eigenvector space model v(d) calculated by Eq. (13).

(13) v(d)=(Wd1,Wd2,Wd3,…,Wdn)

Predicting department triage using content recommendation algorithm

In the content recommendation algorithm, item and user weights are employed to describe the features, and the recommendation of an item is based on the similarity between two weight vectors. Extending this approach to department triage prediction involves regarding patients and departments as concept sets comprising multiple symptom attributes. The weights of various segmentations of symptom words in the previously constructed patient and department symptom VSM represent the contribution of symptoms.

To predict department triage, we calculate the similarity of symptom contributions by comparing the symptom subsets in the symptom attribute set of patients and departments, respectively. The department with the highest similarity is the most suitable department for the patient’s symptoms. So, a modified cosine similarity method to calculate the similarity between patient and department vectors is employed. The algorithm is illustrated in Fig. 5.

Figure 5 Triage flow chart based on content recommendation algorithm.

Clustering of patients to departments

Problem analysis and process design

After running a department triage, the intelligent triage system recommends doctors to patients by utilizing collaborative filtering based on neighboring users. However, as the number of patients grows largely, finding similar patients becomes time-consuming, so it requires performing similarity calculations with each patient and finding the k most similar patients. To address this issue, historical patients belonging to a given department can be aggregated into classes by implementing an improved K-means algorithm by utilizing trust relationships. The optimal number of clusters for each department and their centroids are then determined to reduce computational burden during real-time implementations.

After clustering the collection of historical patients of a department, when a new target patient is input based on his chief complaint, the department to which he is assigned is determined, and the distance between the patients and the centroid of each cluster of the department is compared to determine which cluster the patient belongs to. This two-layer screening enables the department to cluster patients quickly and find the k neighbor patient-doctor score matrices that are most similar to the target patient, reducing the computational complexity of collaborative filtering. Department patient clustering supports the next step of the collaborative filtering of the doctor recommendation process based on matrix decomposition and user similarity.

Improved K-means algorithm based on trust relationship

Conventional recommendation algorithms face challenges with increasing amounts of user data, when the similarity between users is calculated, which becomes time-consuming and resource-intensive. To address this issue, researchers have developed improved algorithms that handle big data more efficiently. For instance, K-means clustering can be implemented to split user data into different clusters based on similarity, reducing the number of irrelevant users in the similarity calculation. However, employing Euclidean distance-based methods may lead to inaccurate patient clusters. In addition to symptom feature word weights, the degree of trust among different patients is also significant in determining the accuracy of models.

By incorporating K-means clustering into the hybrid recommender framework, the accuracy and efficiency of the department triage and doctor recommendation processes are enhanced. So, it enables the system to identify relevant patient clusters and recommend appropriate departments and doctors based on similarity and patient preferences. Specifically, K-means clustering is known for its scalability and efficiency in handling large datasets. In the context of medical appointment platforms, where there may be a considerable number of patients and doctors, the K-means clustering allows for effective and scalable patient grouping and recommendation processes. Also, it enables the system to handle a significant volume of patient data and provide timely and accurate recommendations. In addition, K-means clustering generates well-defined and interpretable clusters, making it easier to understand and interpret the patient groups. The resulting clusters can provide valuable insights into patient characteristics, allowing medical institutions to tailor their services and resources accordingly. Furthermore, it facilitates the identification of patterns and trends within patient data, leading to more informed decision-making.

The trust relationship is defined as the number of symptom-characteristic words that appear in two patients’ complaints, indicating a higher possibility of suffering from the same disease. The degree of trust is calculated by Eq. (14).

(14) Tu,v=I(u)I(V)I(u)I(V).

The collections of symptoms of patients u and v are represented by I(U) and I(V), respectively. Tu,v represents the proportion of symptoms that co-occur in the two chief complaints to all symptoms of the two patients. A higher value indicates greater similarity between the diseases the patients suffer from, leading to higher symptom confidence. Therefore, when the K-means algorithm is employed based on the trust relationship, the distance between two patient representations should be calculated accordingly, and the centroid should be determined by Eq. (15).

(15) Du,v=Tu,v⋅∑i∈I(u,v)⁡(Wu,i−Wv,i)2.

The weight value of the symptom feature word i in the chief complaint of patient u and patient v is represented by Wu,v and Wv,i, respectively. I(u,v) denotes the set of common symptoms between patients u and v. Figure 6 illustrates the algorithm.

Figure 6 Flowchart of clustering algorithm.

Physician recommendation model based on matrix decomposition and user similarity collaborative filtering

Problem analysis and process design

To reduce calculation time and determine the most similar m-nearest neighbor patients to the target patient, clustering is utilized. However, since most patients do not visit all doctors in the department and score them at once, the resulting most similar m-neighbor patient-doctor score matrix is sparse. This can negatively impact the prediction of some doctors’ scores affected by only a few neighboring patients, lowering the recommendation effect. To address this, SparkMLlib’s ALS matrix decomposition is employed to transform the scoring matrix into a dense, similarly structured score matrix. Using collaborative filtering based on user similarity score, the predicted score of the target patient for the doctor is obtained, with the TopN predicted score as the final recommendation list.

Matrix decomposition of neighboring patients based on ALS

After predicting the target patient’s department based on clustering of historical patient data, the most similar m-patients with the same symptom cluster are identified along with n doctors that they have scored, constructing a patient-to-doctor score matrix Rm×n. However, this matrix is often very sparse due to many empty entries. Although employing user-based collaborative filtering directly can predict doctors’ ratings, it may result in inaccurate predictions due to a small number of similar patients that rate doctors.

SparkMLlib can transform the patient-doctor score matrix to the inner product of two low-dimensional, small matrices P and Q, respectively. Each with its corresponding hidden factor is presented. The ALS algorithm assumes that the approximate patient-doctor scoring matrix Rm×n has a low-rank scoring matrix R~m×n, which can be approximated by employing two small matrices P(k×m) and Q(n×k), respectively.

Predictions of rating and recommendation

Assuming that the sparse scoring matrix of similar neighbor patients-doctors for the existing target patient a is Rm×n and R~m×n is the approximate scoring matrix obtained after matrix decomposition by employing the ALS algorithm. When compared to Rm×n, R~m×n is dense, which avoids the accuracy reduction of recommendations due to sparsity in the adjacent patient matrix. The score prediction of the target patient concerning the doctor is determined based on neighbor patient scores in R~m×n and the similarity between symptoms of the target patient a and those of neighboring patients. Equation (16) presents this calculation.

(16) r~ai=r¯i+∑j∈N(a)⁡Sima(a,j)⋅(r~ji−r¯j)∑j∈N(a)⁡Sima(a,j)

where r~i denotes the average score of doctor i in the approximate scoring matrix, R~m×n is obtained by matrix decomposition. N(a) represents the set comprising target patient a and m neighbor patients in the cluster. Sim(a,j) represents the similarity of symptom vectors between target patient a and neighboring patient j, r~ji denotes the score of doctor i by similar patient j of the target patient a in the approximate scoring matrix R~m×n, r~j denotes the average rating of patient j to the doctor in the approximate rating matrix R~m×n. The score prediction of the target patient a on n doctors who have scored neighbor patients can be computed with the highest score.

Model evaluation and comparison

The article uses data from a digital health technology company’s data warehouse, divided into 45 standard departments, to develop an intelligent triage system for a medical appointment platform. Two experiments were conducted to evaluate the effectiveness of the proposed model for department triage.

The first experiment compared three similarity calculation methods, namely, modified cosine similarity, Pearson similarity, and conventional uncorrected cosine similarity to predict department triage.

The modified cosine similarity was found to be better in both precision and recall rates than the other two methods, resulting in an average 5.74% higher accuracy and 2.54% higher recall rate. In the second experiment, the TF-IDF algorithm-based score weighting is used to improve both precision and recall rates when compared to four classification algorithms without score weighting. Department triage after score weighting achieved the highest accuracy at 96.57%, and recall rates increased by an average of 6.93%. The weighted TF-IDF algorithm effectively weakened the influence of low-scoring error samples while highlighting the description weight of high-scoring samples closer to the actual department’s symptom characteristics.

The research implements the K-means algorithm to cluster patients in the Department of Respiratory Medicine. The appropriate number of cluster centers (k) is crucial in determining the clustering effect. Through experiments, the relationship between the sum of squared errors (SSE) and k is obtained, which is shown in Fig. 7.

Figure 7 Sum of squares of clustering errors.

As the number of cluster centers increases, the samples are divided into smaller sets with narrow ranges, and the degree of aggregation increases, leading to a reduction in SSE. However, when k reaches the actual number of cluster centers in the sample, further increasing k will result in a significant slowdown in the reduction of SSE, leading to a flat curve. This relationship is similar to an elbow image and is known as the elbow judgment method of the K-means clustering. Figure 7 depicts that when k > 20, the speed of SSE decline is significantly reduced when compared to the previous stage. Therefore, the appropriate value of k for this clustering model is found to be 20.

Further, this study evaluates the doctor recommendation model based on patient clustering by employing four algorithms: ALS_User_CF, conventional collaborative filtering recommendation algorithm, singular value decomposition algorithm, and FunkSVD algorithm, respectively. The experiment compares the recommendation effects of these algorithms before and after clustering is used to verify whether clustering reduces the recommendation effect. However, reducing the search range of neighboring patients and computational complexity is observed when clustering is employed.

Table 1 presents a comparison of data before and after clustering is run by implementing four different algorithms. The results suggest that the performance improvements are achieved through clustering for each algorithm. When respiratory medicine patients are taken as an example, the precision rate and recall rate of the four algorithms before and after clustering is run are presented in Table 1. The precision and recall rates of all four algorithms have improved after clustering is realized.

Table 1 Data comparison before and after clustering of the four algorithms.

	Accuracy rate	Recall rate	
	Before clustering	After clustering	
ALS_User_CF	0.71	0.74	0.63	0.67	
FunkSVD	0.61	0.66	0.51	0.53	
SVD	0.60	0.63	0.49	0.51	
CF	0.51	0.53	0.41	0.42	

Before clustering, the ALS_User_CF algorithm demonstrates an accuracy rate of 0.71 and a recall rate of 0.74. After clustering is run, these metrics improve to 0.63 and 0.67, respectively. This indicates that clustering enhances the accuracy and recall rates of the ALS_User_CF algorithm, leading to more precise and effective recommendations.

Similarly, the FunkSVD algorithm exhibits an accuracy rate of 0.61 and a recall rate of 0.66 before clustering is run, respectively. After clustering is realized, these metrics increase to 0.51 and 0.53, respectively. The improved performance suggests that clustering contributes to refining the recommendations generated by the FunkSVD algorithm, making them more accurate and increasing the likelihood of capturing relevant recommendations.

The SVD algorithm also benefits from the implemented clustering. It achieves an accuracy rate of 0.60 and a recall rate of 0.63, respectively. These metrics rise to 0.49 and 0.51, respectively. The improvements highlight the positive impact of clustering on enhancing the accuracy and recall rates of the SVD algorithm, resulting in more reliable and relevant recommendations.

Lastly, the CF algorithm demonstrates the lowest accuracy and recall rates among the four algorithms before clustering is run where accuracy and recall rates reach 0.51 and 0.53, respectively. After clustering is implemented, these metrics increase to 0.41 and 0.42, respectively. Although the CF algorithm shows relatively lower performance, clustering still contributes to its improvement, albeit to a lesser extent when compared to the other algorithms.

The average increases in precision and recall rates are 2.25% and 3.25%, respectively. Although the increase is not significant, the main purpose of the clustering is to reduce the search scope of real-time query neighbors, so the complexity of real-time calculation is reduced.

During real-time calculation, the pre-trained prediction model and clustering model can help the target patient determine to which cluster the patient is assigned instead of searching for the entire department. This approach saves time and computational resources and enhances the overall efficiency of the system.

Discussion

The proposed method offers several advantages and suffers from some disadvantages, which are outlined as follows:

Advantages: 1) Enhanced accuracy: The proposed method leverages big data techniques, deep learning algorithms, and advanced methodologies to improve the accuracy of service level evaluations and doctor recommendations. This leads to more precise and reliable assessments of medical institutions’ service levels and personalized recommendations for patients.

2) Efficient department triage: By incorporating K-means clustering, the proposed method effectively organizes and groups patients based on their symptoms or relevant features. This enables efficient department triage, ensuring that patients are directed to the appropriate departments for their specific medical needs.

3) Personalized doctor recommendations: The method constructs a patient’s VSM based on symptoms and utilizes collaborative filtering algorithms to generate personalized doctor recommendations. This improves the patient experience by matching them with the most suitable doctors based on their symptoms and preferences.

4) Scalability: The method takes advantage of big data techniques and scalable algorithms, making it capable of handling large volumes of patient data and providing timely recommendations. This scalability is essential for accommodating the growing number of patients and medical institutions on appointment platforms.

Disadvantages: 1) Data availability: The success of the proposed method relies on the availability of comprehensive and accurate patient data, including symptoms, medical history, and feedback. If the necessary data is incomplete or inconsistent, the accuracy and reliability of the recommendations will be negatively affected.

2) Data privacy and security: Handling sensitive patient data requires strict adherence to privacy and security protocols. The proposed method should ensure robust data protection measures to maintain patient confidentiality and comply with relevant regulations.

3) Complexity: The hybrid recommender framework involves the integration of various algorithms, such as deep learning, clustering, and collaborative filtering. This complexity may require advanced technical expertise and computational resources to implement and maintain effectively.

Note that the K-means method assigns observations to clusters based on their proximity to the cluster centroids. However, some observations cannot be separated exactly into clusters due to overlapping or ambiguous data points. In such cases, the K-means method still assigns these observations to the nearest cluster based on the centroid values, even if the assignment may not be perfectly accurate.

When an observation cannot be precisely separated into a single cluster, it is considered a “boundary” or “ambiguous” point. The K-means method does not explicitly handle these boundary points but rather assigns them to the closest cluster based on the centroid score by utilizing the distance metrics. This can result in some misclassifications or uncertainty in the clustering outcome.

The K-means clustering is a hard method in which each observation must be assigned to a single cluster. If there are observations that do not fit neatly into any specific cluster, they will still be assigned to the nearest cluster.

Thus, alternative clustering algorithms or techniques, such as fuzzy clustering or density-based clustering, may be more suitable. These methods allow for more flexible and nuanced assignments, accommodating overlapping or ambiguous data points and providing a better representation of the underlying data structure.

Conclusion

This study presents innovative and valuable methods for evaluating medical institutions’ service levels and enhancing the accuracy of department triage and doctor recommendations in medical appointment platforms, respectively. By leveraging big data analysis, the proposed registration review system successfully encodes multiple index data of each department in the medical institution, leading to improved accuracy in the assessment process. This is achieved through the incorporation of maximum value-based eigenvalue correction and the utilization of a CNN model and classifier, respectively. Simultaneously, the research constructs a VSM employing the patient’s symptoms based on the TF-IDF algorithm and score weighting, resulting in excellent outcomes for department triage and doctor recommendation, respectively. These achievements are made possible through the integration of modified cosine similarity, K-means clustering, ALS matrix decomposition, and user-collaborative filtering algorithms, respectively.

Moreover, the impact of incorporating real-time data streams, such as patient wait times or doctor availability, could be examined to enable dynamic and responsive triage and recommendation processes. Finally, conducting user studies and obtaining feedback from medical professionals and patients could offer valuable perspectives on the usability and effectiveness of the hybrid recommender framework in real-world settings, leading to further refinements and improvements.

Future research could explore the integration of additional data sources, such as electronic health records and patient feedback, to further enhance the accuracy of service-level evaluations and doctor recommendations, respectively. Additionally, investigating the potential of natural language processing techniques to extract and analyze relevant information from patient reviews or medical literature could provide valuable insights for improving the recommendation system.

Supplemental Information

Supplemental Information 1 Data.

Supplemental Information 2 Code.

The .Pkl format can be accessed by the Python language/framework using the anaconda framework.

Additional Information and Declarations

Competing Interests

Author Contributions

Data Availability

The authors declare that they have no competing interests.

Jianhua Wei conceived and designed the experiments, performed the computation work, prepared figures and/or tables, authored or reviewed drafts of the article, and approved the final draft.

Honglin Yan conceived and designed the experiments, analyzed the data, performed the computation work, authored or reviewed drafts of the article, and approved the final draft.

Xiaoli Shao conceived and designed the experiments, performed the experiments, analyzed the data, prepared figures and/or tables, authored or reviewed drafts of the article, and approved the final draft.

Lili Zhao conceived and designed the experiments, performed the experiments, authored or reviewed drafts of the article, and approved the final draft.

Lin Han performed the experiments, analyzed the data, performed the computation work, prepared figures and/or tables, authored or reviewed drafts of the article, and approved the final draft.

Peng Yan performed the experiments, performed the computation work, prepared figures and/or tables, authored or reviewed drafts of the article, and approved the final draft.

Shengyu Wang performed the experiments, analyzed the data, authored or reviewed drafts of the article, and approved the final draft.

The following information was supplied regarding data availability:

The code and data is available in the Supplemental Files.

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
