# Peer review of "A machine learning-based hybrid recommender framework for smart medical systems"

_PeerJ Computer Science, doi:10.7717/peerj-cs.1880_

## Round 0.1 · original submission · Major Revisions

Dear Authors,
Thank you for your submission to our esteemed journal. I think your paper needs some major points to consider for improvement according to the suggestions of the experts.
Therefore, we are sending you the experts' comments for your consideration. further please consider the following points as well
1. Improve the technical language of your manuscript.
2. justify the novelty of your work compared to existing literature.
3. who are the potential beneficiaries of your work?

**Language Note:** The review process has identified that the English language must be improved. PeerJ can provide language editing services - please contact us at copyediting@peerj.com for pricing (be sure to provide your manuscript number and title). Alternatively, you should make your own arrangements to improve the language quality and provide details in your response letter. – PeerJ Staff

Reviewer 1 ·

Basic reporting

The manuscript contributes to the literature with a solid foundation. However, the issues detected should be either improved or corrected.
1. The abstract should be shorter. It should be rewritten and reorganized.
2. The number of articles discussed in the introduction is very limited. It should be increased. More discussion should be added to the introduction section.
3. All equations should be cited in the text. Instead of using the word “formula”, Eq. (.), should be utilized.
4. Proofreading is a must.
5. Paragraphs are very long. They should be shortened. The references section has several different reference types. They need to be checked and fixed.

Experimental design

1. Why did the authors not use fuzzy c-means clustering since several clusters are interrelated when medical practices are a concern? Please discuss it.
2. Did the authors check the existence of influential observations or outliers since they affect the cluster's center and the clusters' observations? Please discuss it.
3. “Degree of Trust” is a subjective measurement. Did the authors test how this kind of subjectivity affects the results? Please discuss.
4. The proposed method should be presented in an algorithm that presents the steps.

Validity of the findings

1. The conclusion section should be rewritten and reorganized. The current form is poor.
2. Since data was multi-model data, did the authors conduct any preprocessing steps? Please discuss it.
3. Why did the author pick 4 methods to compare? How did they decide them? Verify it with the literature.

Cite this review as

Reviewer 2 ·

Basic reporting

The article has some severe technical issues that need to be fixed. Also, the template and the language should be taken care of carefully. A major revision is needed.
1. When three similarity measures are compared, did the authors run any statistical test to decide which one is better, or a mathematical comparison is run? Please discuss.
2. Both the discussion and conclusion sections should be reconstructed.
3. What kind of pooling is run to reduce feature dimensions? Please discuss.
4. How was the CNN implemented? For example, what are convolution dimensions, pooling layer, hidden layer numbers activation function type, and so on? Please discuss them.
5. The proposed method should be presented in a different section with more explanations and remarks.
6. Did the authors run any preprocessing steps to normalize multi-scale vectors? Please discuss it.
7. What are the percentages of training, test, and validation data sets?
8. This sentence is extracted from the text: “Although the increase is not significant, the main purpose of the clustering algorithm is to reduce the search scope of real-time query neighbors to reduce the complexity of real-time calculation.” How did the author decide that the increase is not significant? Did they run any analysis to determine that the search scope of the real-time query decreases the complexity of real-time calculations? If done where are the available results? If not, please remove those statements.
9. Instead of using SSE in Figure 7, MSE (means square error) should be used to decide which k (cluster number) is better since the average is more sensitive than the summation.
10. Why did authors prefer to use mean for cluster over median since the median is robust statistics?
11. Even though one of the research subjects is “a doctor recommendation”, a solid discussion is presented in the article. Please check and add more remarks.

Experimental design

What kind of pooling is run to reduce feature dimensions? Please discuss

Validity of the findings

What are the percentages of training, test, and validation data sets?

Cite this review as

---

## Round 0.2 · accepted · Accept

Based on the input from the experts, I'm pleased to notify you that your manuscript is scientifically suitable for publication. Congratulations

Reviewer 1 ·

Basic reporting

The article is improved now and satisfy all important requirements to be published and preceding for further steps.

Experimental design

All the comments regarding experimental design are improved now and it satisfy all basic concerns.

Validity of the findings

Now the article is meeting the validating criteria for the findings and those are well contributing in the relevant research community.

Cite this review as

Reviewer 2 ·

Basic reporting

Clear

Experimental design

Enough

Validity of the findings

Ok

Cite this review as